# The full-length BEND2 protein is dispensable for spermatogenesis but required for setting the ovarian reserve in mice

Yan Huang[1,2], Nina Bucevic[1,2], Carmen Coves[1,2], Natalia Felipe-Medina[3], Marina Marcet-Ortega[1,2], Nikoleta Nikou[1,2], Cristina Madrid-Sandín[1,2], Maria López-Panadés[1,2], Carolina Buza[1,2], Neus Ferrer Miralles[4,5,6], Antoni Iborra[7], Anna Pujol[8], Alberto M Pendás[3], Ignasi Roig[1,2]*

[1]Genome Integrity and Instability Group, Institut de Biotecnologia i Biomedicina, Universitat Autònoma de Barcelona, Cerdanyola del Vallès, Spain; [2]Department of Cell Biology, Physiology, and Immunology, Cytology and Histology Unit, Universitat Autònoma de Barcelona, Cerdanyola del Vallès, Spain; [3]Molecular Mechanisms Program, Centro de Investigación del Cáncer and Instituto de Biología Molecular y Celular del Cáncer (CSIC-Universidad de Salamanca), Salamanca, Spain; [4]Institut de Biotecnologia i de Biomedicina, Universitat Autònoma de Barcelona, Cerdanyola del Vallés, Spain; [5]Centro de Investigación Biomédica en Red de Bioingeniería, Biomateriales y Nanomedicina, Instituto de Salud Carlos III, Cerdanyola del Vallès, Spain; [6]Departament de Genètica i de Microbiologia, Universitat Autònoma de Barcelona, Cerdanyola del Vallés, Spain; [7]Servei de Cultius Cel·lulars, Producció d'Anticossos i Citometria, Universitat Autònoma de Barcelona, Cerdanyola del Vallés, Spain; [8]Transgenic Animal Unit, Center of Animal Biotechnology and Gene Therapy, Universitat Autònoma de Barcelona, Cerdanyola del Vallés, Spain

*For correspondence:
ignasi.roig@uab.cat

## eLife Assessment

This study provides **valuable** information on a novel gene that regulates meiotic progression in both male and female meiosis. The evidence supporting the conclusions of the authors is **solid**. This study will be of interest to developmental and reproductive biologists.

**Abstract** Infertility affects up to 12% of couples globally, with genetic factors contributing to nearly half of the cases. Advances in genomic technologies have led to the discovery of genes like *Bend2*, which play a crucial role in gametogenesis. In the testis, *Bend2* expresses two protein isoforms: full-length and a smaller one. Ablation of both proteins results in an arrested spermatogenesis. Because the *Bend2* locus is on the X chromosome, and the *Bend2*[-/y] mutants are sterile, BEND2's role in oogenesis remained elusive. In this study, we employed a novel *Bend2* mutation that blocks the expression of the full-length BEND2 protein but allows the expression of the smaller BEND2 isoform. Interestingly, this mutation does not confer male sterility and mildly affects spermatogenesis. Thus, it allowed us to study the role of BEND2 in oogenesis. Our findings demonstrate that full-length BEND2 is dispensable for male fertility, and its ablation leads to a reduced establishment of the ovarian reserve. These results reveal a critical role for full-length BEND2 in oogenesis and provide insights into the mechanisms underlying the establishment of the

ovarian reserve. Furthermore, these findings hold relevance for the diagnostic landscape of human infertility.

## Introduction

Infertility is a complex and multifaceted issue affecting approximately 8–12% of couples worldwide, with genetic factors playing a significant role in up to 50% of cases (*Ding and Schimenti, 2021*). The intricate process of human reproduction involves a delicate interplay of numerous genes and biological pathways, making the genetic landscape of infertility both diverse and challenging to unravel.

Advancements in genomic technologies and high-throughput sequencing have facilitated the identification of novel genes implicated in reproductive processes. Among these discoveries, *Bend2* has recently emerged as a novel gene involved in mammalian gametogenesis (*Malcher et al., 2022*). BEND2 is a member of the BEN domain-containing protein family, whose members play diverse roles in cellular processes such as transcriptional regulation, chromatin remodeling, and protein-protein interactions (*Abhiman et al., 2008*).

Mouse models have been instrumental in identifying novel genes involved in mammalian fertility. This approach has significantly expanded our understanding of the genetic factors underlying reproductive processes in both males and females (*Garretson et al., 2023*). A recent study reported that BEND2 is required to complete spermatogenesis since it is a crucial regulator of mammalian meiosis in spermatocytes (*Ma et al., 2022*). Click or tap here to enter text. BEND2 is specifically expressed in spermatogenic cells shortly before and during the prophase of meiosis I. In the testis, *Bend2* expresses two protein isoforms: full-length, named p140, and smaller one, p80. BEND2 is essential for the transition from zygonema to pachynema, as its knockout in male mice arrests meiosis at this stage. Its absence leads to disrupted synapsis and induced non-homologous chromosomal pairing. BEND2 interacts with multiple chromatin-associated proteins, including components of transcription-repressor complexes. Additionally, BEND2 regulates chromatin accessibility and modifies H3K4me3, influencing the overall chromatin state during spermatogenesis. These functions collectively establish BEND2 as a key regulator of meiosis, gene expression, and chromatin state in mouse male germ cells. Nonetheless, its role in oogenesis remains unexplored since the *Bend2* locus is on the X chromosome, and mutant males are sterile.

In this study, we aimed to elucidate the role of *Bend2* in oogenesis using a novel mutation of *Bend2* that does not confer male sterility. Using this novel *Bend2* mutation, we have investigated the effects of full-length BEND2 ablation on meiosis, oocyte quality, and follicular dynamics. Furthermore, as this mutation does not cause male infertility, it allowed us to investigate the role of BEND2 in mammalian meiosis in more detail. Our findings contribute to expanding our understanding of the genetic factors influencing mammalian fertility and provide potential avenues for diagnosing genetic determinants of human infertility.

## Results

### BEND2 is highly expressed in spermatogenic nuclei before meiosis initiation and during early meiotic prophase I, independent of meiotic recombination

Taking advantage of several unannotated transcripts detected in 14 dpp wild-type mouse testis from our previous RNA sequencing analysis (*Marcet-Ortega and Roig, 2016*), we found a novel splice variant of the then annotated *Gm15262* gene, which is an X-linked gene containing two BEN domains (*Figure 1A*). This variant is specifically expressed in mouse testes and fetal ovaries containing germ cells undergoing meiotic prophase I and shows a nuclear localization at the heterochromatin of spermatocytes when the GFP-tagged splice variant is electroporated to the testes of young mice (*Figure 1—figure supplement 1A–B*). Thus, we hypothesize that this gene could be essential for spermatocyte and oocyte development. Based on sequence homology, we named this *Gm15262* gene as the mouse homolog of the human *Bend2*, and thus, we renamed it *Bend2*, as also proposed by others recently (*Ma et al., 2022*).

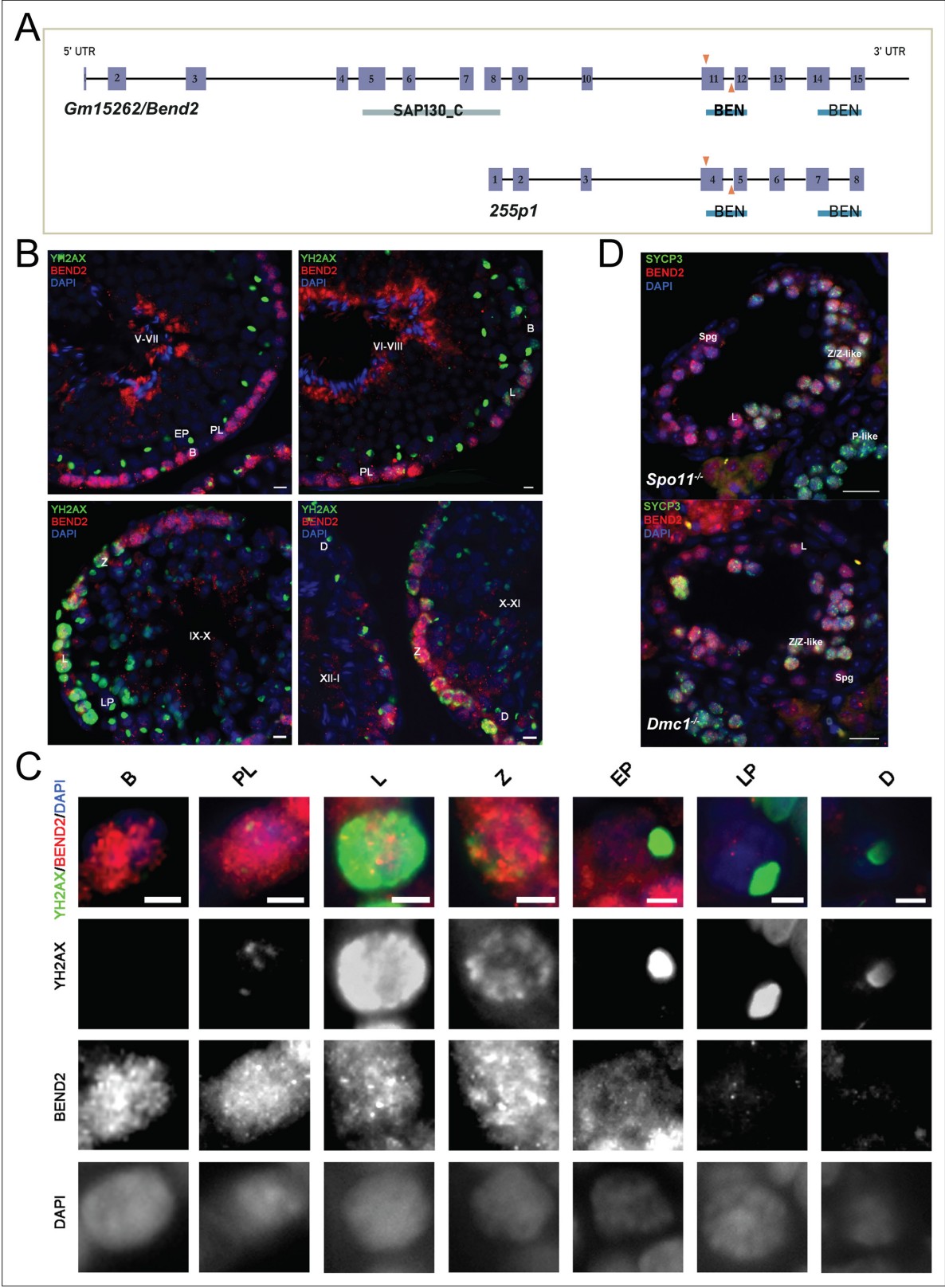

**Figure 1.** Expression of BEND2 during spermatogenesis. (**A**) Schematic representation of mouse *Gm15262/Bend2* and its novel splice variant *255* p1. The exons are shown as purple boxes. The predicted domains are labeled below the exons. SAP130_C: C-terminal domain of histone deacetylase complex subunit SAP130; BEN: BEN domain. A pair of gRNAs (orange arrows) target exon 11 of *Bend2* within the first BEN domain to generate the *Bend2*$^{D11}$ mutation by CRISPR/Cas9. The *in-house* BEND2 antibody is generated using full-length 255P1 sequence as immunogen. (**B**) BEND2

*Figure 1 continued on next page*

*Figure 1 continued*

localization in wild-type mouse testis. Staging of the seminiferous epithelium is based on the localization of spermatocytes (indicated by the expression and localization of YH2AX (green)) and spermatids (indicated by DAPI). The tubule stage is shown in uppercase Roman numerals. Scale bar, 20 µm. (C) Magnification of BEND2-positive cells from testis sections. Expression of BEND2 was characterized in cells along spermatogenesis from spermatogonia to late diplotene spermatocyte. Scale bar, 10 µm. (D) BEND2 localization in SPO1- and DMC1-deficient testis. In these cases, SYCP3 (green) was used to identify spermatocytes. Scale bar, 50 µm. Spg: spermatogonia; B: B-type spermatogonia; PL: pre-leptotene spermatocyte; L: leptotene spermatocyte; Z: zygotene spermatocyte; Z-like: zygotene-like spermatocyte; EP: early pachytene spermatocyte; LP: late pachytene spermatocyte; P-like: pachytene-like spermatocyte; D: diplotene spermatocyte.

The online version of this article includes the following source data and figure supplement(s) for figure 1:

**Figure supplement 1.** Expression and localization of *Bend2* (**A**) Expression of 255P1 in mouse gonads and somatic tissues by RT-PCR.

**Figure supplement 1—source data 1.** Source images for *Figure 1—figure supplement 1A*.

**Figure supplement 1—source data 2.** Source images for *Figure 1—figure supplement 1A*.

We generated a polyclonal antibody against BEND2 using the novel splice variant of *Bend2* as an immunogen (see validation in *Figure 2*). To examine BEND2 expression during spermatogenesis, we immunostained PFA-fixed testis sections against BEND2 and the DNA damage marker γH2AX to allow the identification of spermatocytes. BEND2 was abundantly detected in the nuclei of the peripheral cells of tubules from stage V until XII-I (*Figure 1B*). This nuclear staining first appeared in spermatogonia before DSBs were generated. BEND2 staining was highly present in spermatocytes from pre-leptotene when meiosis initiates and persisted until early pachytene. In late pachytene and diplotene cells, little BEND2 remained in the nuclei of spermatocytes (*Figure 1C*). Notably, this expression pattern of BEND2 is reminiscent of the one reported before (*Ma et al., 2022*).

Furthermore, to reveal if the localization of BEND2 in germ cells could have originated as a response to critical events of the meiotic prophase, we examined BEND2 expression in testis sections from recombination-defective mice lacking SPO11. In SPO11-deficient testis, where no DSBs are formed at the onset of meiosis, spermatocytes cannot initiate meiotic recombination, thereby failing to synapse and entering apoptosis (*Barchi et al., 2005*; *Baudat et al., 2000*; *Pacheco et al., 2015*). Interestingly, we found BEND2 was extensively present in nuclei of spermatogonia, leptotene, zygotene, and zygotene-like spermatocytes, as in wild-type mice (*Figure 1D*). Very occasionally, a few BEND2 signals could be observed in some SPO11-deficient spermatocytes, presumably at a more advanced pachytene-like stage. This is consistent with its reduced expression after early pachytene in wild-type mice. Similarly, BEND2 expression was not altered in *Dmc1* mutant spermatocytes (*Figure 1D*), which cannot complete meiotic recombination (*Barchi et al., 2005*; *Pittman et al., 1998*). Altogether, these results indicate that although BEND2 is highly expressed during early meiotic prophase I when meiotic recombination initiates and progresses, its expression starts as early as in spermatogonia before meiosis begins and is independent of either DSB formation or completion of recombination.

We also examined BEND2 expression in 16 dpc fetal ovary sections by IF and found it localizing at nuclei of some zygotene stage oocytes (*Figure 1—figure supplement 1C*). However, due to scarce samples and technical obstacles, we could not characterize BEND2 expression in oocytes at earlier stages from 12 to 14 dpc ovary sections.

## Full-length BEND2 deficiency causes increased apoptosis and persistent unrepaired DSBs in spermatocytes

To address the germ cell functions of BEND2 in mice, we generated BEND2-deficient mice by CRISPR/Cas9. Part of exon 11 of *Bend2* was targeted to be removed to disrupt the first BEN domain (*Figure 1A*). *Bend2Δ11/y* mice developed into adults without apparent differences in general physical appearance compared to their littermates. Male *Bend2D11/y* mice were fertile. The size and weight of *Bend2D11/y* testes were comparable to that of wild-type testes (*Figure 2A* and *Supplementary file 1*). Similarly to *Ma et al., 2022*, using our BEND2 antibody, we detected two BEND2-related proteins in wild-type testes by WB: one was ~130 kDa, and another was ~75 kDa (*Figure 2C*). Although the predicted size of full-length BEND2 is 80.9 kDa, based on previous reports (*Ma et al., 2022*), the 130 kDa protein corresponds to the full-length BEND2, which displays a slower electrophoretic mobility due to its unusual sequence/structure. The 75 kDa protein corresponds to a smaller version of BEND2 lacking the N-terminus, p80 (*Ma et al., 2022*). Interestingly, we did not detect any BEND2

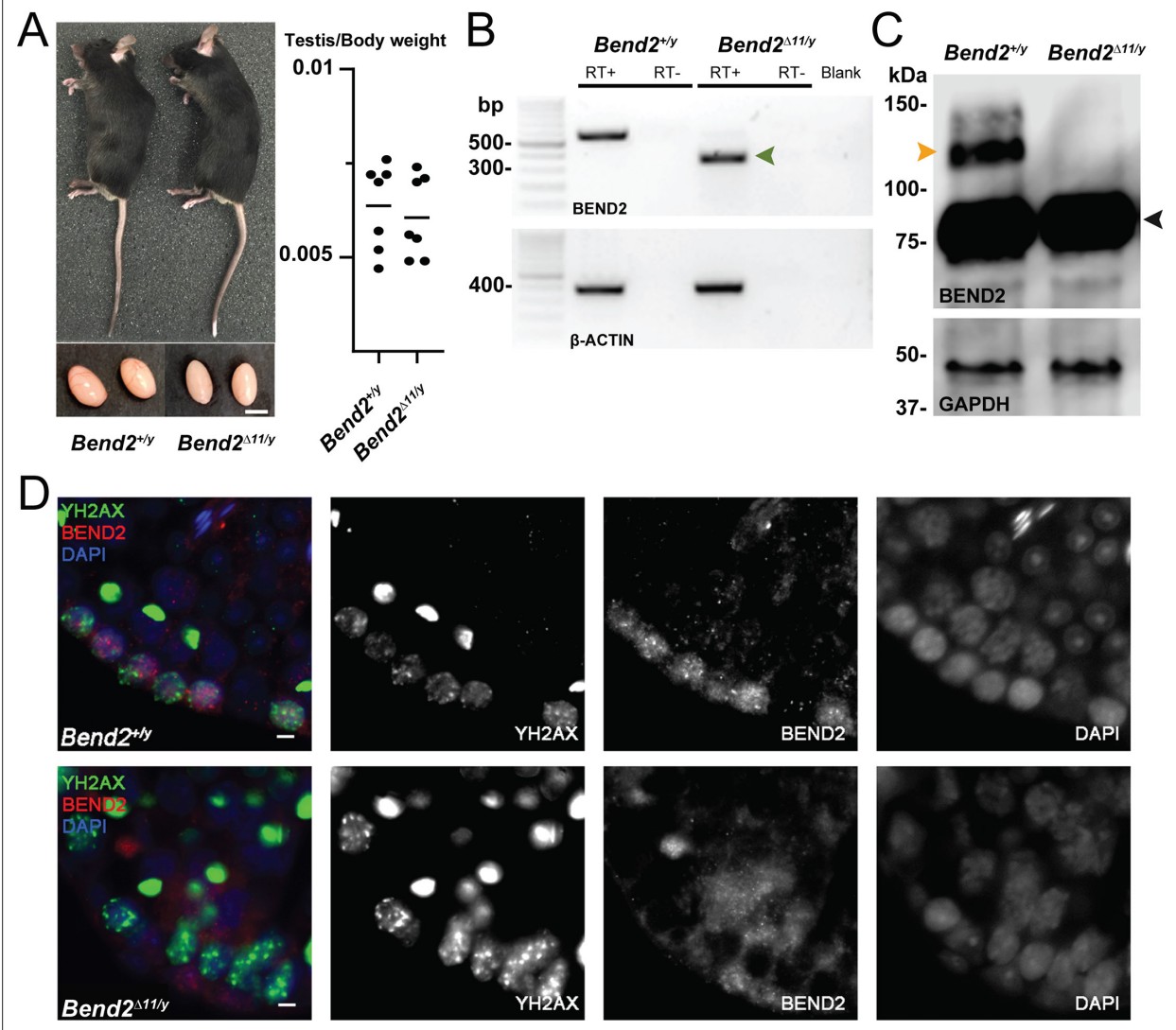

**Figure 2.** Disruption of BEND2 in *Bend2^{D11/y}* mice. (**A**) Mouse appearance and testis size. Scale bar, 5 mm. (**B**) *Bend2* expression in testis by RT-PCR. The expected size of the wild-type (WT) allele was 576 bp. Sanger sequencing results showed that the amplified DNA in the mutated allele (372 bp, green arrowhead) was from an mRNA that resulted from skipping exon 11 of *Bend2*. (**C**) Detection of BEND2 by WB from testis protein extracts. The orange arrowhead indicates the full-length BEND2 protein band. An extra protein band ~75 kDa (black arrowhead) was also detected in both wild-type and mutant testes. (**D**) Detection of BEND2 by IF. Testis sections were treated with antigen retrieval using Tris-EDTA buffer before staining against BEND2 (red) and SYCP3 (green). The BEND2 staining observed in the *Bend2* mutants closely resembles background staining (not shown), suggesting that the cytoplasmic signal might be nonspecific. Scale bar, 10 μm.

The online version of this article includes the following source data for figure 2:

**Source data 1.** Source images for *Figure 2B*.

**Source data 2.** Source images for *Figure 2B*.

protein in wild-type testes of 48.3 kDa, which would be expected from the novel transcript 255p1 we identified.

We were unable to detect the full-length BEND2 in *Bend2^{D11/y}* mouse testis extracts by WB (*Figure 2C*). The staining of BEND2 observed in wild-type cells was not present in mutant cells (*Figure 2D*). However, the p80 BEND2-related protein was still present in *Bend2^{D11/y}* mutant testes (*Figure 2C*). Moreover, an alternative *Bend2* transcript skipping exon 11 was detected in *Bend2^{D11/y}* testis by RT-PCR and sequencing (*Figure 2B*).

We attempted to test if this exon 11-skipped transcript was also present in wild-type mouse testis by RT-PCR and if it represented the p80 BEND2-related protein detected in both wild-type and

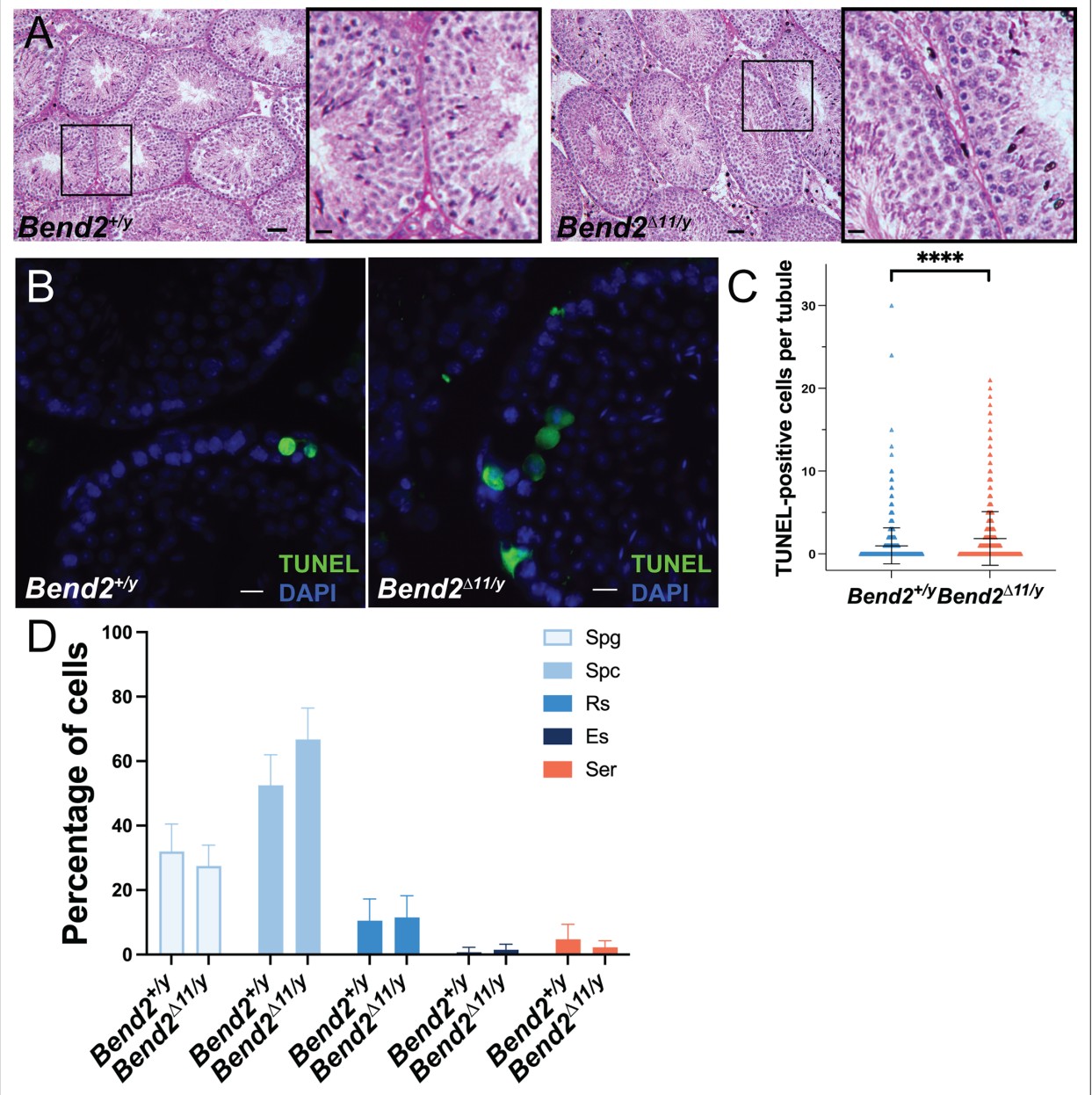

**Figure 3.** The full-length BEND2 is dispensable to complete spermatogenesis in mice. (**A**) Representative PAS-H stained mouse testis sections. The square in the left image shows the zoomed images on the right. Scale bar on the right images, 20 µm. (**B**) Apoptosis detection on testis sections by TUNEL assay. Scale bar, 20 µm. (**C**) Quantification of TUNEL-positive cells. The horizontal lines represent the mean ± SD. N=1,125 for *Bend2^{+/y}*; N=1195 for *Bend2^{D11/y}*, ****p<0.0001 t-test. (**D**) Classification of TUNEL-positive cells. The columns and error lines indicate the mean ± SD. N=4, p>0.05 One-Way ANOVA.

*Bend2^{D11/y}* testis by immunoprecipitation (IP) coupled to peptide mass fingerprinting analysis. Unfortunately, only the full-length *Bend2*-202 transcript was consistently amplified in our RT-PCR analysis, and our *in-house* BEND2 antibody did not give satisfactory IP results (*Data was not shown*). Overall, these results suggest that the full-length BEND2 was successfully eliminated from mutant mouse testis. Also, the p80 BEND2 protein, likely corresponding to an exon 11, skipped transcript, is present and might be functional in our mutant testis, based on the observed phenotype (see below).

Histological analysis of *Bend2^{D11/y}* testes revealed the presence of cells at all the stages of spermatogenesis (***Figure 3A***). However, a significant increase in apoptotic cells was detected in *Bend2^{D11/y}* testes via TUNEL assays (p<0.0001 t-test, ***Figure 3B–C***). Further staging analysis showed that *Bend2^{D11/y}*

mice presented a comparable number of apoptotic spermatocytes (66.8 ± 9.7%, mean ± SD, N=4) than wild-type mice (52.5 ± 9.5%, mean ± SD, N=4, p>0.05 One-Way ANOVA, *Figure 3D*). Therefore, the full-length BEND2 might be dispensable for completing spermatogenesis, and its absence only caused a slight defect in spermatogenesis.

To further explore the causes of the increased apoptosis in BEND2–deficient testes, we assessed chromosome synapsis and meiotic recombination progress in spermatocytes by IF. In wild-type spermatocytes, SYCP3 began to form each developing chromosome axis at leptotene. At zygotene, synapsis is initiated as SYCP1 appears at the synapsed region of the homologs. At pachytene, synapsis was completed as SYCP3 and SYCP1 completely colocalized. At diplotene, SCs disassembled, and SYCP1 was lost from separated SYCP3-labelled axes, but homologous chromosomes remained held together by chiasmata.

Synapsis progressed similarly to wild-type spermatocytes in *Bend2$^{D11/y}$* spermatocytes (*Figure 4A*). However, the fraction of diplotene spermatocytes was significantly increased (p=0.019, One-Way ANOVA). Additionally, the fraction of pachytene spermatocytes (48.6 ± 6.8%, mean ± SD, N=4) seemed to decrease in *Bend2$^{D11/y}$* mice, compared to wild-type mice (57.3 ± 4.5%, mean ± SD, N=4, p=0.078 One-Way ANOVA). This accumulation of *Bend2$^{D11/y}$* spermatocytes at the diplotene stage could be explained if there was a later block of the meiotic progression or if *Bend2$^{D11/y}$* spermatocytes might exit pachytene faster than wild-type spermatocytes. Taken together, these results demonstrated that in the absence of the full-length BEND2, spermatocytes could complete synapsis and progress through meiotic prophase but following a slightly altered timeline.

During early meiosis, recombination initiates with SPO11-mediated DSB formation, leading to the phosphorylation of the histone H2AX by ATM and thus triggering a series of DSB repair responses in the meiotic prophase (*Huang and Roig, 2023*). In wild-type mice, ϒH2AX progressively disappeared from the autosomes while DSBs were repaired as prophase progressed. By pachytene, most ϒH2AX was associated with the sex body. Few ϒH2AX patches could be observed on the autosomes from late pachytene, presumably corresponding to unrepaired DSBs (*Figure 4B*).

Interestingly, a marked increase in the number of ϒH2AX patches was detected from early pachytene until late diplotene in *Bend2$^{D11/y}$* spermatocytes, indicating a possible delay in DSB repair (*Figure 4B*). Alternatively, the same results could be observed if more DSBs were formed during prophase in *Bend2$^{D11/y}$* spermatocytes.

Once DSBs are resected, replication protein A (RPA) transiently binds to the nascent ssDNA overhangs, promoting the assembly of the recombinases RAD51 and DMC1 at DSB sites (*Moens et al., 2002*). RAD51 and DMC1 replace RPA and form nucleoprotein filaments with the ssDNA, directing homology search and strand invasion to proceed along the DSB repair pathway (*Brown and Bishop, 2015*; *Hinch et al., 2020*).

In wild-type spermatocytes, the number of RPA foci peaked around early zygotene and progressively diminished as recombination proceeded; on the other hand, there were many RAD51 foci since leptotene, and the number started to decline from early zygotene (*Figure 4C–D*), consistent with previous studies (*Moens et al., 2002*; *Pacheco et al., 2015*). In *Bend2$^{D11/y}$* males, the spermatocytes had similar initial RPA and RAD51 foci numbers, suggesting that DSB formation was not affected by the loss of BEND2 (*Figure 4C–D*). However, *Bend2$^{D11/y}$* cells tended to accumulate more RPA foci and lose RAD51 foci as they progressed along the meiotic prophase. The differences became significant for RPA at late leptotene (p=0.0015 t-test, *Figure 4C*) and RAD51 at late zygotene and pachytene (p=0.0428 and 0.0002, respectively t-test, *Figure 4D*), indicating a subtle defect in the replacement of RPA by RAD51.

Because of meiotic recombination, at least one crossover per bivalent is generated to ensure accurate chromosome segregation at the first meiotic division (*Zickler and Kleckner, 1999*). We examined MLH1 foci, which become apparent at mid-late pachytene and mark most crossover-designated sites (*Anderson et al., 1999*). We did not detect significant differences in the number of MLH1 foci in *Bend2$^{D11/y}$* spermatocytes (22.0 ± 2.3, mean ± SD, N=68) compared to wild-type cells (22.5 ± 2.9, mean ± SD, N=74, p=0.2543, t-test, *Figure 4E*), suggesting that BEND2 deficiency did not affect crossover formation in spermatocytes.

Homologous recombination is the principal DSB repair pathway in meiosis responding to SPO11-dependent DSBs. But in late spermatocytes (late pachytene and diplotene), other DSB repair pathways, such as non-homologous end joining (NHEJ) or inter-sister homologous recombination (IS-HR),

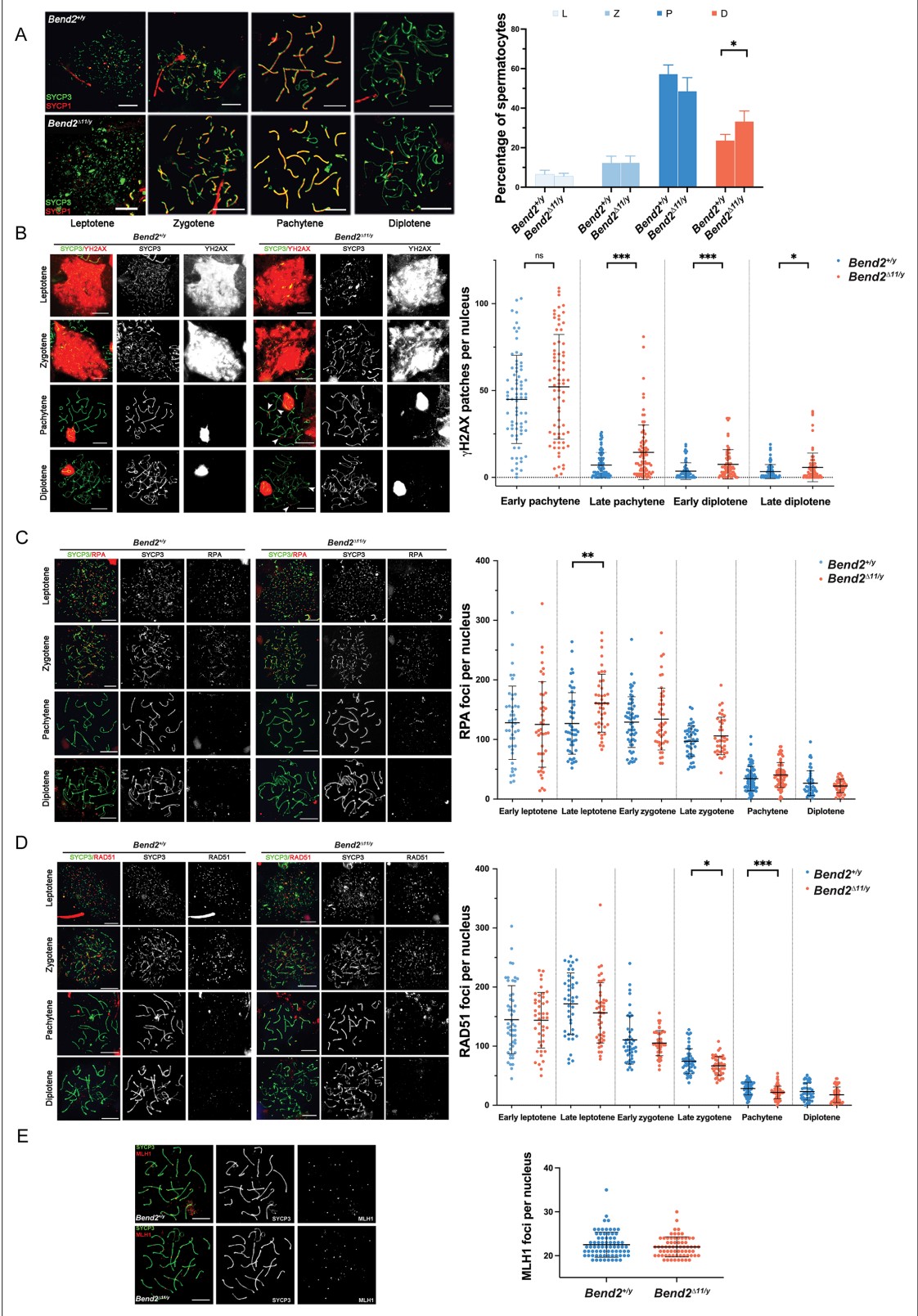

**Figure 4.** *Bend2^{D11/y}* males display minor recombination defects. (**A**) Chromosomal synapsis in spermatocytes. Representative images of SYCP3 and SYCP1 staining in spermatocyte nuclei from the stages shown (left). Meiotic prophase staging of spermatocytes (right). L: leptotene, Z: zygotene, P: pachytene, D: diplotene. The columns and lines indicate the mean and SD. N=4, *p=0.019 One-Way ANOVA. (**B**) Examination of DSBs in *Bend2^{D11/y}* spermatocytes. Representative images of YH2AX staining in spermatocyte nuclei along meiotic prophase (left). Increased YH2AX signals are detected

*Figure 4 continued on next page*

Figure 4 continued

during late prophase in *Bend2^D11/y* spermatocyte (white arrowhead). Quantification of ɤH2AX patches in spermatocyte nuclei per sub-stages (right). The patches were counted manually using the same method for every pair of control and mutant mice. From left to right, N=76/74, 86/83, 73/75, and 78/81, ***p=0.0001 (late pachytene) and 0.0007 (early diplotene), *p=0.0222 t-test. Analysis of replication protein A (RPA) (**C**) and RAD51 (**D**) foci counts in control and *Bend2^D11/y* spermatocytes. RPA and RAD51 foci were counted using ImageJ with the same method for every *Bend2^+/y* and *Bend2^D11/y* mice pair. From left to right, N=44/43, 51/43, 59/46, 45/39, 81/77, and 48/43, **p=0.0015 for RPA quantification; N=52/43, 42/46, 44/46, 51/45, 77/73, and 45/49, *p=0.0428 and ***p=0.0002 for RAD51 quantification; t-test. (**C**) Examination of CO formation in *Bend2^D11/y* spermatocytes. Representative images of MLH1 in control and mutant spermatocyte nuclei (left). Quantification of MLH1 foci in spermatocyte nuclei (right). Only spermatocytes containing ≥19 MLH1 foci/nucleus were counted. N=74 for *Bend2^+/y* and 68 for *Bend2^D11/y*, p>0.05 Mann-Whitney test. The horizontal lines represent mean ± SD (**B–E**). Scale bar, 10 µm.

The online version of this article includes the following source data and figure supplement(s) for figure 4:

**Figure supplement 1.** Examination of the non-homologous end joining (NHEJ) pathway activity in *Bend2^D11/y* mice.

**Figure supplement 1—source data 1.** Source images for *Figure 4D*.

**Figure supplement 1—source data 2.** Source images for *Figure 4D*.

**Figure supplement 2.** Analysis of LINE-1 retrotransposon expression in *Bend2^D11/y* mice.

take over (*Ahmed et al., 2010*; *Enguita-Marruedo et al., 2019*; *Goedecke et al., 1999*). Moreover, during pachytene, new DSBs could be formed by SPO11-independent mechanisms (*Carofiglio et al., 2013*). Therefore, the increased presence of ɤH2AX patches observed in *Bend2* mutant mice might be explained by the inactivation of these alternative DSB repair pathways during the late meiotic prophase or an increase of SPO11-independent DSBs during pachytene.

To gain insight into these hypotheses, we studied if the NHEJ pathway activity is reduced during meiosis in *Bend2^D11/y* mice by examining the localization and expression of its constitutive protein, Ku70, on testes sections. As anticipated, Ku70 was observed as a patch-like signal on sex bodies in pachytene and diplotene spermatocytes. A punctuated signal, marking presumably DSB sites in the nucleus, was seen in cells of all spermatogenesis stages besides round and elongated spermatids (*Figure 4—figure supplement 1A*). The number of Ku70 foci colocalizing with SYCP3 in spermatocytes in pachytene and diplotene stages was similar between control and mutant samples (*Figure 4—figure supplement 1B*). No differences were observed in the proportion of tubules expressing Ku70 either. These results were further confirmed with Western blot analysis (*Figure 4—figure supplement 1C–D*), thus suggesting that *Bend2* depletion does not affect NHEJ activation.

## BEND2 deficiency leads to LINE-1 retrotransposon suppression

The formation of SPO11-independent DSBs during meiotic prophase has been linked to the LINE-1 retrotransposon activation (*Carofiglio et al., 2013*; *Malki et al., 2014*; *Soper et al., 2008*). To evaluate if the LINE-1 activity was increased in *Bend2* mutant mice, we examined LINE1 expression in mouse testes by IF and WB.

As expected, the most prominent LINE-1 staining localized to the cytoplasm of leptotene, zygotene, and pachytene spermatocytes (*Figure 4—figure supplement 2A*). The signal intensity varied greatly between seminiferous tubules of the same sample in both wild-type and mutant animals. In *Bend2^D11/y* mice, spermatocytes showed similar signal intensity compared to control mice. Contrary to the expected, a statistically significant lower number of LINE-1 positive tubules was observed in mutant mice. An average of 2.8% and 0.5% positive tubules was observed in control and *Bend2^D11/y* mice, respectively (p=0.02, t-test) (*Figure 4—figure supplement 2B*). Western blot analysis validated the lower LINE-1 protein expression in *Bend2^D11/y* testes sections (*Figure 4—figure supplement 1B*). Indeed, a significant decrease in LINE-1 protein levels was found in *Bend2^D11/y* mice (p=0.0469, t-test, *Figure 4—figure supplement 2C*), thus confirming the IF results. Defects in the demethylation efficiency of LINE-1 in *Bend2* mutant mice during the prophase may explain these results. On the other hand, the signal intensity was equal in wild-type and mutant mice, suggesting that some tubules successfully complete LINE-1 demethylation, thus resulting in normal LINE-1 expression.

## BEND2 deficiency results in a reduced ovarian reserve

A more robust phenotype was observed in females when BEND2 was disrupted compared to males. Female *Bend2^D11/D11* mice were fertile. However, the litter size was significantly smaller than that

in wild-type females (*Figure 5A*), and noticeably smaller ovaries were present in mutant females (*Figure 5B*).

To investigate if the loss of BEND2 affected oogenesis, a more detailed histological analysis of whole ovaries was performed in young (1-week-old), adolescent (3-weeks-old), adult (15–20-weeks-old), and aged (40-weeks-old) *Bend2*^D11/D11^ and *Bend2*^+/+^ mice. At 1 week, *Bend2*^D11/D11^ females exhibited a significantly reduced number of oocytes (p=0.0012, t-test; *Figure 5C*), mainly due to a reduced number of primordial follicles (p=0.0007, t-test; *Figure 5—figure supplement 1A*). Growing follicles were found at all stages, with no statistically significant differences between *Bend2*^D11/D11^ and control mice (*Figure 5—figure supplement 1A*). This tendency continued in prepubertal 3-week-old animals. At this stage, Bend2 mutant ovaries seemed to contain fewer oocytes (p=0.0834, t-test; *Figure 5C*) and primordial follicles (p=0.0704, t-test; *Figure 5—figure supplement 1B*). At an adult stage of 15–20 weeks, the mutant ovaries presented significantly fewer oocytes (p=0.0040, t-test; *Figure 5C*), again due to a reduction in the number of primordial follicles (p=0.0024, t-test; *Figure 5—figure supplement 1C*). Interestingly, at this stage, we also noticed a significant reduction in the number of antral follicles in mutant mice (p<0.0001, t-test; *Figure 5—figure supplement 1C*), which could partly explain the reduced fertility of these females. Finally, at 40 weeks, there tended to be fewer oocytes and primordial follicles in *Bend2*^D11/D11^ ovaries than in *Bend2*^+/+^ ovaries (p=0.1171 and p=0.1383, t-test; *Figure 5—figure supplement 1D*). These results suggest a reduction in the oocyte pool established at birth and the possible occurrence of premature ovarian insufficiency in *Bend2* mutant females.

To certify that BEND2 was required to establish the ovarian reserve, we counted the oocyte population present in newborn *Bend2*^+/+^ and *Bend2*^D11/D11^ mutant mice by staining ovarian sections with the germ-cell marker DDX4 (*Figure 5B*). *Bend2* mutant ovaries contained almost 40% fewer oocytes than control ovaries (p=0.0037, t-test; *Figure 5C*), evidencing that the absence of the full-length BEND2 resulted in a reduced ovarian reserve.

## Mutant *Bend2* oocytes show persistent unrepaired DSBs and impaired crossover formation

To address the cause of this reduced ovarian reserve, we first analyze the expression of LINE-1, which has been associated with increased perinatal oocyte death in mice (*Malki et al., 2019*). However, we found a similar percentage of oocytes expressing LINE-1 in *Bend2*^+/+^ and *Bend2*^D11/D11^ ovarian sections (20.2±0.6; 16.8±2.3, mean ± SEM; p=0.1851, t-test; *Figure 5—figure supplement 2*). These data suggest that the reduction in the number of oocytes present in *Bend2* mutant mice might not be related to an increased expression of LINE-1.

Next, we examined if errors during the meiotic prophase could be responsible for the observed reduced ovarian reserve in *Bend2*^D11/D11^ mice. Thus, we studied chromosome synapsis and meiotic recombination progress in fetal oocytes by IF. In wild-type ovaries, most oocytes at 18 dpc were at the pachytene stage, a small fraction was still at the zygotene stage, and a minority at the leptotene stage. At 1 dpp, almost all oocytes were at the pachytene and diplotene stages (*Figure 6A*). In *Bend2*^D11/D11^ ovaries, zygotene, and pachytene oocytes undergoing normal synapsis and diplotene oocytes undergoing desynapsis were observed at 18 dpc and 1 dpp (*Figure 6A*).

Consistent with the observation in *Bend2* mutant males, we also found increased levels of ϒH2AX in late prophase *Bend2*^D11/D11^ oocytes (*Figure 6B*). In wild-type females, ϒH2AX patches could be detected along axes in most pachytene oocytes and many diplotene oocytes (*Figure 6B*). This is different from wild-type males, in which ϒH2AX can only be detected as very few patches in late pachytene cells and is almost undetectable in most diplotene cells (*Figure 4B*), presumably because synapsis progresses faster than recombination in females than in males (*Roig et al., 2004*).

In *Bend2*^D11/D11^ oocytes, higher levels of ϒH2AX were observed at all stages analyzed (early pachytene, late pachytene, and diplotene, *Figure 6B*). Many early pachytene *Bend2*^D11/D11^ oocytes exhibited an overall intense ϒH2AX signal reminiscent of wild-type zygotene cells (*Figure 6B*). To quantitatively compare these differences, we measured the ϒH2AX signal intensity in early pachytene oocytes and counted the ϒH2AX patches in late pachytene and diplotene oocytes. As expected, a significant increase of ϒH2AX was present in *Bend2*^D11/D11^ oocytes at all the tested stages (p=0.0288 for early pachytene; p<0.0001 for late pachytene, early diplotene and late diplotene, t-test, *Figure 6B*). Thus, like in *Bend2* mutant males, DSB repair was impaired, or more DSBs were formed in *Bend2*^D11/D11^ females.

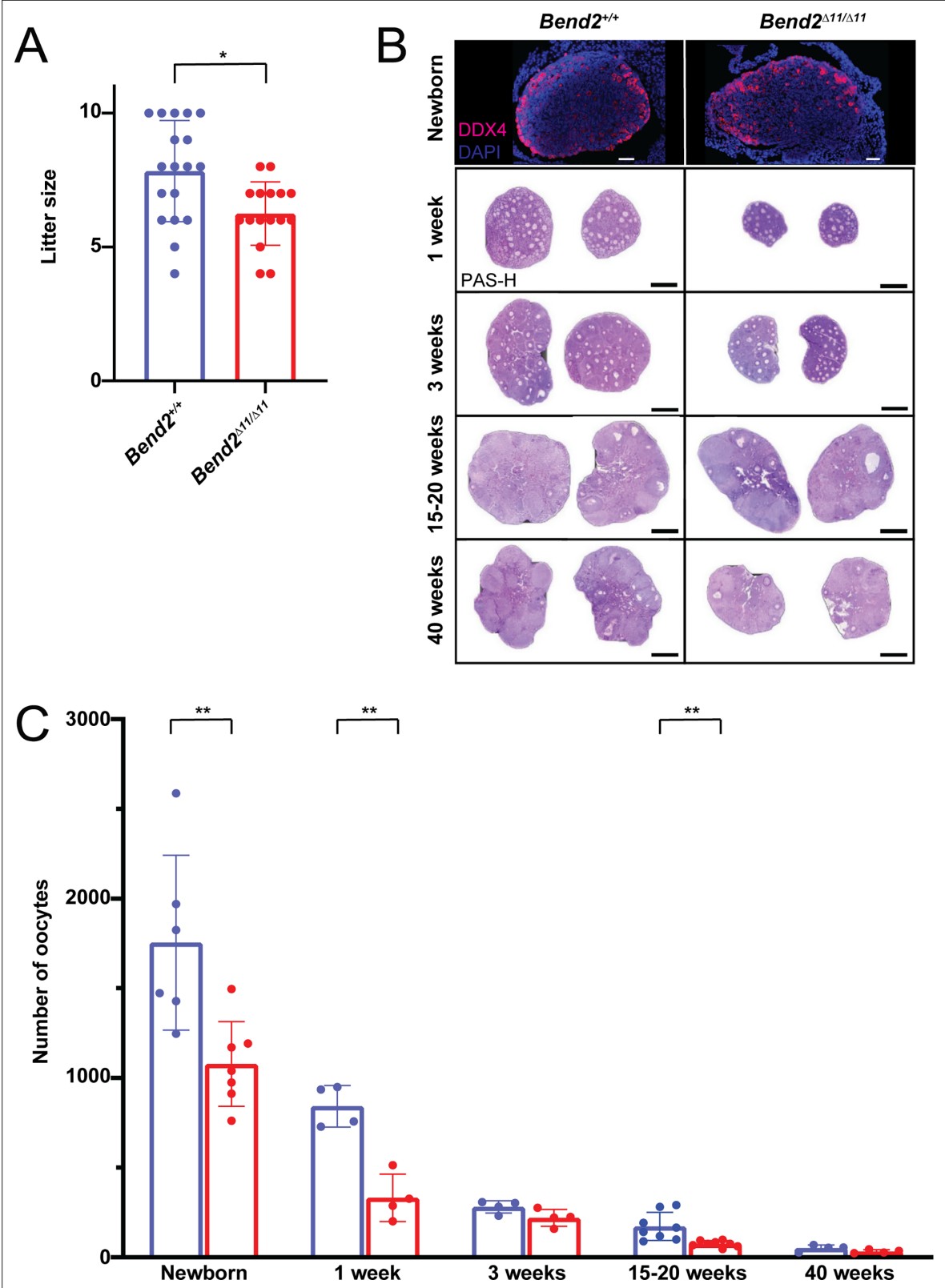

**Figure 5.** Oogenesis is altered in *Bend2* mutant females. (**A**) Fertility evaluation of *Bend2^{D11/D11}* females. Two-month-old *Bend2^{D11/D11}* females were crossed with wild-type males for 5 months; litter size data of four animals per genotype were collected for analysis. *p=0.0069, t-test. (**B**) DDX4-stained and Periodic Acid Schiff-Hematoxylin (PAS-H) stained histological ovary sections from females at different ages. Scale bars, 50 µm for Immunofluorescent

*Figure 5 continued on next page*

*Figure 5 continued*

images and 0.5 mm for the PAS-H stained ones. (**C**) Quantification of the number of oocytes found in the analyzed ovaries of newborn, one, three, 15–20, and 40 weeks old mice. The columns and lines indicate the mean and SD. \*\*p=0.0075, 0.0012, and 0.0040; t-test.

The online version of this article includes the following figure supplement(s) for figure 5:

**Figure supplement 1.** Classification of the type of follicles found in control and *Bend2^{D11/D11}* ovaries.

**Figure supplement 2.** Analysis of LINE-1 retrotransposon expression in *Bend2^{D11/D11}* mice.

In females, no differences in the number of RPA and RAD51 foci were found between wild-type and *Bend2^{D11/D11}* oocytes, apart from a slight decrease in RPA foci in *Bend2^{D11/D11}* oocytes (*Figure 6C*) and a slight increase in RAD51 foci in pachytene *Bend2^{D11/D11}* oocytes (*Figure 6D*). These results suggest that the increased presence of ϒH2AX in *Bend2^{D11/D11}* oocytes may not be related to defective meiotic recombination but to other causes, like meiotic silencing of unsynapsed chromosomes (MSUC) or alternative DNA repair pathways.

Interestingly, the number of MLH1 foci in late prophase oocytes was significantly lower in *Bend2^{D11/D11}* females than in control females (p=0.006 for *Bend2^{D11/D11}*, t-test, *Figure 6E*). Thus, we concluded that the depletion of BEND2 caused a reduced crossover formation in oocytes, probably partly causing the reduced litter size in *Bend2* mutant females.

Finally, to check if these alterations affected *Bend2^{D11/D11}* oocyte quality, we performed ovarian stimulation on 7-month-old *Bend2^{+/D11}* and *Bend2^{D11/D11}* mice. On average, we obtained a similar number of oocytes from the ovarian stimulation in control and mutant mice (10.7±4.1 vs 10.0±1.2; mean ± SEM; p=0.8852, t-test; *Table 1*). Remarkably, the ability of *Bend2^{D11/D11}* mutant oocytes to be fertilized and develop into the blastocyst stage did not seem to be compromised (*Table 1*), suggesting that the absence of the full-length BEND2 did not compromise oocyte quality, at least to be fertilized and develop up to the blastocyst stage.

## Discussion

In mammals, the establishment and gradual depletion of the ovarian reserve defines a finite female reproductive life span. In this study, we demonstrated that BEND2, a new player in meiosis (*Ma et al., 2022*), is essential for the primordial follicle pool setup. The depletion of the full-length BEND2 causes severe defects in female oogenesis with a decreased ovarian reserve and subfertility. However, these defects may not be directly related to meiotic recombination progression or synapsis, although a high level of unrepaired DSBs persists during late meiotic prophase. We also showed that BEND2 has a role in regulating LINE-1 retrotransposon activity in spermatocytes. Our work expands our knowledge of the genetic factors influencing the ovarian reserve, potentially benefiting the genetic diagnosis of human infertility.

BEND2 has two C-terminal BEN domains, marked by α-helical structure, and is conserved in a range of metazoan and viral proteins. BEN domain is suggested to mediate protein–DNA and protein-protein interactions during chromatin organization and transcription (*Abhiman et al., 2008*). So far, only a few BEN domain-containing proteins have been described. All of their functions are linked to transcriptional repression regardless of whether the proteins contain other characterized domains, including mammalian BANP/SMAR1, NAC1, BEND3 and RBB and *Drosophila* mod(mdg4), BEND1, and BEND5 (*Dai et al., 2015*; *Dai et al., 2013*; *Gerasimova et al., 1995*; *Kaul-Ghanekar et al., 2004*; *Korutla et al., 2007*; *Korutla et al., 2005*; *Nègre et al., 2010*; *Rampalli et al., 2005*; *Sathyan et al., 2011*; *Xuan et al., 2013*). BEND1, BEND5, and RBB are revealed to bind DNA through sequence-specific recognition (*Dai et al., 2015*; *Dai et al., 2013*; *Xuan et al., 2013*), and BEND3 specifically binds heterochromatin (*Sathyan et al., 2011*). Additionally, several human BEND2 fusion proteins have been identified in tumors, and the activation of BEND2 is suggested to promote oncogenic activity (*Burford et al., 2018*; *Scarpa et al., 2017*; *Sturm et al., 2016*; *Williamson et al., 2019*). These data suggest that BEND2 might have a role in regulating chromatin and transcriptional gene expression. A recent study showed that BEND2 is a crucial regulator of meiosis gene expression and chromatin state during mouse spermatogenesis (*Ma et al., 2022*).

In the study of *Ma et al., 2022*, two BEND2-related proteins (p140 and p80) were found in mouse testis, using an antibody against a short polypeptide from the C-terminus of BEND2. They

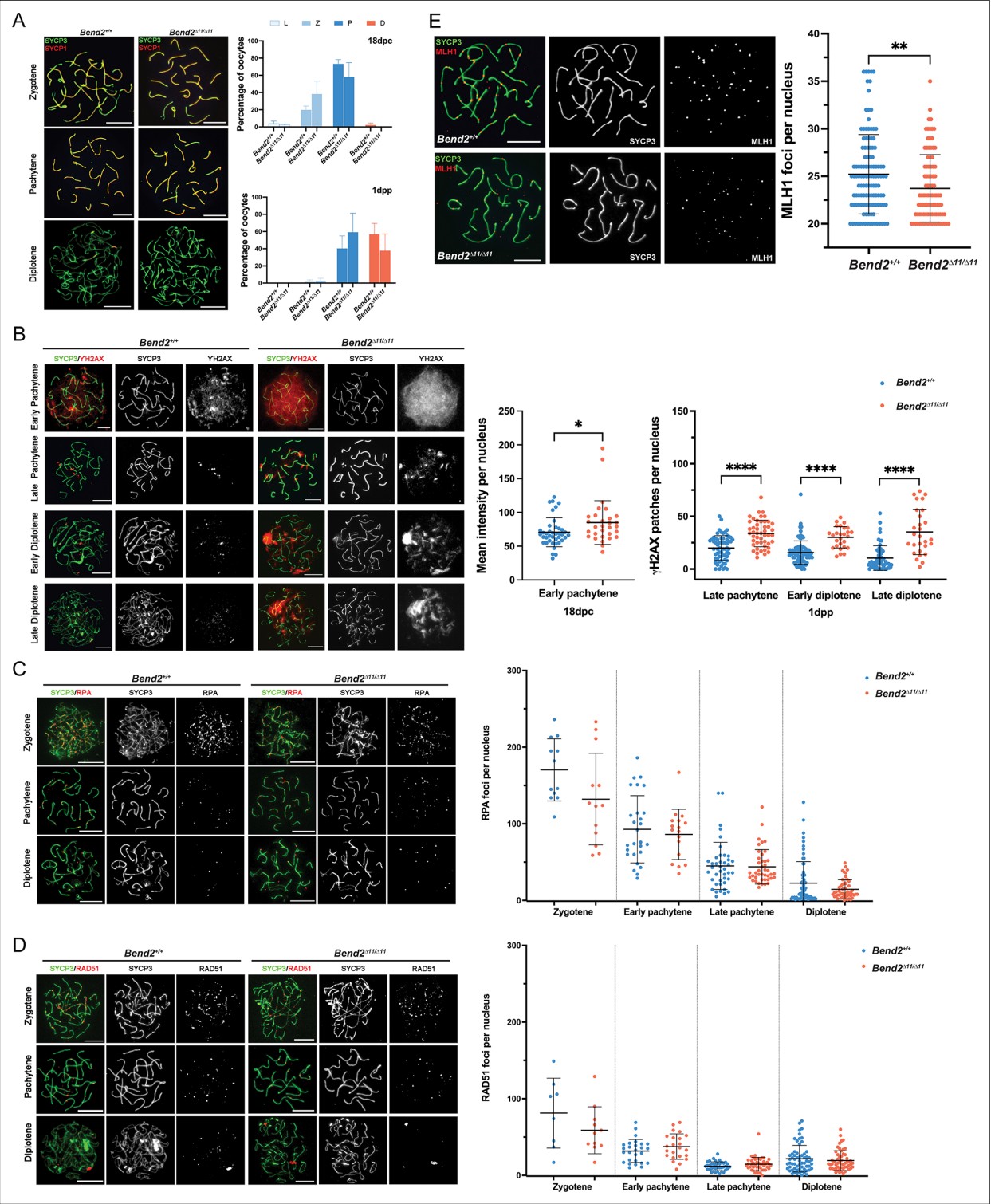

**Figure 6.** Synapsis and recombination in *Bend2^{D11/D11}* females. (**A**) Chromosomal synapsis in oocytes. Representative images of SYCP3 and SYCP1 staining in 18 dpc and 1 dpp oocyte nuclei are shown (left). Meiotic prophase staging of 18 dpc and 1 dpp oocytes (right). L: leptotene, Z: zygotene, P: pachytene, D: diplotene. The columns and lines indicate the mean and SD. The number of animals analyzed per genotype, 2 for 18 dpc and 3 for 1 dpp. p>0.5 for all the comparisons One-Way ANOVA. (**B**) Examination of DSBs in *Bend2^{D11/D11}* oocytes. Representative images of ϒH2AX staining in 18 dpc and 1 dpp oocyte nuclei at the pachytene and diplotene stage (left). Quantification of ϒH2AX patches in oocyte nuclei at sub-stages (right). The mean intensity of ϒH2AX staining in 18 dpc oocyte nuclei was measured by Image J. The patches of ϒH2AX in 1 dpp oocyte nuclei were counted manually. The horizontal lines represent the mean ± SD. From left to right, the number of analyzed nuclei per genotype: 40/30, 64/48, 81/27, and 57/28.

*Figure 6 continued on next page*

*Figure 6 continued*

*p=0.0288, ****p<0.0001 t-test. (**C**) Analysis of replication protein A (RPA) (**C**) and RAD5 (**D**) foci in control and *Bend2*$^{D11/D11}$ oocytes. Representative images of RPA staining in 18 dpc and 1 dpp oocyte nuclei from zygotene to diplotene stage (left). Quantification of RPA foci present in oocyte nuclei at sub-stages (right). Zygotene and early pachytene nuclei analyzed were from 18 dpc ovaries, and late pachytene and diplotene nuclei analyzed were from 1 dpp ovaries. Foci were counted manually. The horizontal lines represent the mean ± SD. From left to right, number of analyzed nuclei: 12/13, 27/17, 41/42, and 65/49 for RPA, and 8/11, 27/23, 38/46, and 62/55 for RAD51. p>0.5 for all the comparisons t-test. (**E**) Examination of CO formation in control and *Bend2*$^{D11/D11}$ oocytes. Representative images of MLH1 in 1 dpp oocyte nuclei (left). Quantification of MLH1 foci in 1 dpp oocyte nuclei (right). Only oocytes containing ≥20 MLH1 foci/nucleus were counted. The horizontal lines represent the mean ± SD. N=120 for *Bend2*$^{+/+}$, 95 for *Bend2*$^{D11/D11}$; **p=0.0057 Mann-Whitney test. Scale bar, 10 μm.

demonstrated that p140 is the full-length version of BEND2, displaying slower electrophoretic mobility due to its unusual sequence/structure at the N-terminus. They also reported that BEND2 was expressed in the nuclei of spermatogenic cells around meiosis initiation. Similarly, using our antibody against the whole C-terminus of the protein, we also found two BEND2-related proteins expressed in mouse testis and a highly consistent expression pattern of BEND2 during spermatogenesis. These two proteins likely correspond to full-length and smaller isoform from *Ma et al., 2022*.

*Ma et al., 2022* disrupted *Bend2* by deleting exons 12 and 13, which resulted in the loss of both the full-length (p140) and the smaller isoform (p80) of BEND2. In their study, *Bend2* mutant mice were sterile and exhibited severe meiotic recombination and synapsis defects. In contrast, our study targeted exon 11, specifically ablating the full-length p140 isoform while preserving the expression of the smaller p80 isoform in the testis. Interestingly, *Bend2*$^{Δ11/y}$ mutant males remain fertile, with essentially normal meiotic recombination and synapsis, suggesting that the full-length BEND2 (p140) is not essential for the repair of SPO11-induced double-strand breaks or for synapsis during meiosis.

This observation strongly implies that p80 is sufficient to fulfill the essential roles of BEND2 in meiosis and spermatogenesis. The preserved fertility and proper progression of meiosis in our *Bend2*$^{Δ11/y}$ mutants suggest that the smaller p80 isoform is the primary driver of BEND2's function in these processes. Conversely, the sterility and severe meiotic defects observed in the study by Ma et al. can be attributed to the loss of p80 and p140. These findings highlight a functional redundancy or compensatory mechanism provided by p80 that allows meiosis and spermatogenesis to proceed effectively in the absence of the full-length isoform.

Thus, we propose that p80 is the indispensable isoform of BEND2 required for meiotic progression. Its presence in our model likely prevents the spermatogenesis arrest seen in the Ma et al. study. This hypothesis provides a compelling explanation for why our *Bend2*$^{Δ11/y}$ males exhibit a less severe phenotype and remain fertile despite the absence of full-length BEND2. These results underscore the importance of considering isoform-specific functions when investigating the roles of multi-isoform genes in complex biological processes like meiosis and gametogenesis.

In our study, one common defect in mutant spermatocytes and oocytes was a significant increase in unrepaired DSBs during late prophase. Examination of the chromosomal dynamics and meiotic recombination demonstrated that the full-length BEND2 is not essential for SPO11-induced DSB repair. Since all meiotic recombination markers studied, apart from gH2AX, were mostly indistinguishable in mutant and control cells, we hypothesized that the observed unrepaired DSBs might be SPO11-independent and further explored the LINE-1 retrotransposon activation, which is known to cause DSBs during meiosis (*Soper et al., 2008*; *Carofiglio et al., 2013*; *Malki et al., 2014*) and it has been implicated in the regulation of the ovarian reserve (*Martínez-Marchal et al., 2020*; *Tharp et al., 2020*). Unexpectedly, we found a significant reduction in LINE-1 expression levels in mutant testis and although it was not statistically significant, mutant oocytes seemed to have a similar trend.

Repetitive transposon elements, including LINE-1, occupy almost half of the human genome and are extremely important for genome evolution (*Wang, 2017*). Transposon-mediated insertional mutagenesis was linked to a multitude of genetic diseases. Even though LINE-1 is active mainly during the genome-wide demethylation, transient DNA demethylation occurring at the onset of meiosis was found to trigger a persisting LINE-1 expression until mid-pachytene (*Soper et al., 2008*). Silencing of LINE-1 retrotransposons is achieved with multiple mechanisms such as DNA methylation, histone modifications, and piRNAs (piwi-interacting RNAs) (*Di Giacomo et al., 2013*). *Ma et al., 2022* described that in *Bend2* KO male mice, there is an up-regulation of PIWIL2 expression at the onset of meiosis. PIWIL2 belongs to the PIWI subfamily of proteins

**Table 1.** Bend2<sup>D11/D11</sup> oocytes fertilize and develop into blastocyst stage similar to control oocytes.

| Group | Genotype | # mouse | Age (months) | # oocyte | # 2 cell embryo | Fertilization (%) | Blastocyst (%) |
|---|---|---|---|---|---|---|---|
| Control | Bend2<sup>+/D11</sup> | 3 | 7 | 10.7±7.2 | 7.7±3.8 | 78.0±19.5 | 23.3±25.2 |
| Mutant | Bend2<sup>D11/D11</sup> | 3 | 7 | 10.0±2.0 | 8.3±3.2 | 82.7±20.5 | 46.8±12.2 |

that bind to piRNAs, which are required for several biological processes, including silencing retrotransposons during meiosis (*Di Giacomo et al., 2013*). Thus, it is likely that *Bend2* depletion caused an up-regulation of the piRNA pathway, resulting in lower LINE-1 expression in our mouse mutant spermatocytes.

Alternatively, as BEND2 was found to bind to repetitive sequences (*Ma et al., 2022*), it could interact directly with LINE-1 to promote its expression. The lower LINE-1 expression might result straightforwardly from less binding of BEND2. Finally, *Bend2* mutation could affect chromatin state and higher-order structures (*Ma et al., 2022*), which could alter LINE-1 expression and hinder DNA repair. Further studies focusing on the regulation of PIWIL2 by BEND2 will clarify these aspects of the phenotype.

Due to the sterility of hemizygous mutant males, female homozygous mutants can't be produced in the study of *Ma et al., 2022*. So, our female mutant model facilitates further exploration of BEND2's roles in oogenesis.

Interestingly, our mutant females *Bend2<sup>D11/D11</sup>* exhibited a more severe phenotype with a reduced ovarian reserve and subfertility, which had not been described before. Mouse females establish the pool of primordial follicles around birth. As folliculogenesis begins, the pool is gradually depleted throughout reproductive life (*Hunter, 2017*). By examining the dynamics of the oocyte pool in female mice at different ages, we showed that compared to wild-type females, BEND2-deficient females have a significantly smaller ovarian reserve. As a result, fewer follicles develop during folliculogenesis from pre-puberty throughout adulthood. Consistent with these, BEND2-deficient females produced reduced litter size, which indicates subfertility. Interestingly, our mutant oocyte quality analysis suggests that mature oocytes from mutant females are equally competent to develop into a blastocyst as control ones. These data suggest that the subfertility observed in *Bend2* mutants may be due to errors in later developmental stages, such as implantation or organogenesis.

Thus, BEND2, apart from its roles in male meiosis (*Ma et al., 2022*), is also required to establish the ovarian reserve during oogenesis. The full-length BEND2 may take on this role in females. However, future investigation of BEND2 expression in fetal ovaries is necessary to validate this. Genes discovered to have disease-causing roles in mouse models often offer a panel of strong candidate genes for screening human infertility factors (*Huang and Roig, 2023*; *Riera-Escamilla et al., 2019*). Our data shows that the depletion of BEND2 may lead to premature ovarian insufficiency (POI), which is also a significant cause of female infertility in humans, and its underlying genetic causes are mainly unknown (*Rossetti et al., 2017*). Thus, identifying BEND2's requirement for normal oogenesis is also important to recognizing the genetic determinants of human POI.

# Materials and methods

## Key resources table

| Reagent type (species) or resource | Designation | Source or reference | Identifiers | Additional information |
|---|---|---|---|---|
| Gene (*M. musculus*) | Bend2 | Ensembl | ENSMUSG00000108981 | *Gm15262* |
| Genetic reagent (*M. musculus*) | *Bend2<sup>D11/D11</sup>* or *Bend2<sup>D11/y</sup>* | This paper | | Available from the authors upon request Dr. Ignasi Roig Ignasi.Roig@uab.cat |
| Sequence-based reagent | gRNA 1-antisense | This paper (Sigma) | gRNA | AGTAGCAGGCTGCATAAGT GGG |

*Continued on next page*

*Continued*

| Reagent type (species) or resource | Designation | Source or reference | Identifiers | Additional information |
|---|---|---|---|---|
| Sequence-based reagent | gRNA 2-sense | This paper (Sigma) | gRNA | AGACCAGCCTTATTGACCA TGG |
| Strain, strain background (*Escherichia coli*) | 5-alpha competent *E. coli* cells | New England Biolabs. | C2987H | competent cells |
| Strain, strain background (*Escherichia coli*) | *E. coli* BL21(DE3) | NCBI | 469008 | competent cells |
| Cell line (*H. sapiens*) | HEK 293T | Cellosaurus | CVCL_0063 | Cell line maintained in SCAC, Institut de Biotecnologia i de Biomedicina |
| Antibody | anti-BEND2 Rb | This paper | | IF(1:100), WB (3.2 µg/ml) Available from the authors upon request Dr. Ignasi Roig Ignasi.Roig@uab.cat |
| Antibody | anti-Ku70 Rb | Abcam | ab92450 | IF (1:100) WB (1:2000) |
| Antibody | anti-GAPDH Rb | Abcam | ab37168 | WB (1:2000) |
| Antibody | anti- SYCP3 Ms or Rb | Abcam | ab97672 or ab15093 | IF (1:200) |
| Antibody | anti- SYCP1 Rb | Abcam | ab15090 | IF (1:200) |
| Antibody | anti-phospho-Histone H2A.X Ms | Millipore | 05–636 | IF (1:400) |
| Antibody | anti-MLH1 Ms | BD Biosciences | 51-1327GR | IF (1:50) |
| Antibody | RPA32 (4E4) Rat | Cell Signalling | 2208 S | IF (1:100) |
| Antibody | anti-RAD51 (ab-1) Rb | Millipore | PC130 | IF (1:100) |
| Antibody | anti-GFP Rb | Thermo Fisher Scientific | A-11122 | IF (1:200) |
| Antibody | anti-LINE1 ORF1p Rb | Abcam | ab216324 | IF (1:100) WB (1:2000) |
| Antibody | anti-rabbit HPR | Bio-Rad | 170–6515 | WB (1:5000) |
| Recombinant DNA reagent | pEGFP-C1 | Clontech | Catalog:6084–1 | |
| Recombinant DNA reagent | pEGFP-N1 | Clontech | Catalog:6085–1 | |
| Recombinant DNA reagent | | This paper | | Available from the authors upon request Dr. Ignasi Roig Ignasi.Roig@uab.cat |
| Recombinant DNA reagent | pEGFP-N1-BEND2 | This paper | | Available from the authors upon request Dr. Ignasi Roig Ignasi.Roig@uab.cat |
| Recombinant DNA reagent | pET-28a (+)-TEV | This paper | | Dr. Neus Ferrer Miralles Institut de Biotecnologia i de Biomedicina, |
| Sequence-based reagent | forward | This paper | PCR primers | BEND2 genotyping; TTGCCAGTGGGGTATTACGA |
| Sequence-based reagent | reverse | This paper | PCR primers | BEND2 genotyping; CAGGGCATTTGCACCCCATGCC |
| Sequence-based reagent | forward | This paper | PCR primers | BEND2 female genotyping; TTTGCTCCACTGTTTCACGC |
| Sequence-based reagent | reverse | This paper | PCR primers | BEND2 female genotyping; TCCCTTTAAACTGCCAACAACA |
| Sequence-based reagent | 255P1 forward | This paper | PCR primers | 255P1 Gene specific primers for RT-PCR; TAGGGACCAAGAACCTGCTG |

*Continued on next page*

*Continued*

| Reagent type (species) or resource | Designation | Source or reference | Identifiers | Additional information |
|---|---|---|---|---|
| Sequence-based reagent | reverse | This paper | PCR primers | 255P1 Gene specific primers for RT-PCR; TCCTGAAGCCACTGAGAAGG |
| Sequence-based reagent | forward | This paper | PCR primers | β-actin for RT-PCR; AGGTCTTTACGGATGTCAACG |
| Sequence-based reagent | reverse | This paper | PCR primers | β-actin primers for RT-PCR; ATCTACGAGGGCTATGCTCTC |
| Sequence-based reagent | forward | This paper | PCR primers | BEND2 cloning primer; agaaATGCCAGGAAAAACTGAAG |
| Sequence-based reagent | reverse | This paper | PCR primers | BEND2 cloning primer; TTAAGCTATTGCATTCCTTGGG for pEGFP-C1 vector;gAGCTATTGCATTCCTTGGGC for pEGFP-N1 vectors |
| Peptide, recombinant protein | His-tagged BEND2 protein | This paper | | Available from the authors upon request Dr. Ignasi Roig Ignasi.Roig@uab.cat |
| Commercial assay or kit | jetPEI DNA transfection reagent | Polyplus Transfection | 101–40 N | |
| Commercial assay or kit | HisTrap HP column | Cytiva | 29051021 or 17524801 | |
| Commercial assay or kit | TUNEL reaction mixture | Roche Diagnostics | 11684795910 | *Figure 3* |
| Software, algorithm | Image J | Image J | RRID:SCR_003070 | |
| Software, algorithm | GraphPad Prism 8 | GraphPad | RRID:SCR_002798 | |
| Software, algorithm | Adobe Photoshop | Adobe | RRID:SCR_014199 | |
| Other | DAPI stain | Invitrogen | D1306 | 0.1 µg/ml |
| Other | Vectashield antifade mounting medium | Vector Laboratories | H1000 | |

## Mice

*Bend2* mutant mice were generated by using the CRISPR/Cas9 system. A pair of gRNAs with minimum off-target and maximum on-target activity were designed and selected to specifically target sequences that encode essential protein domains of the *Bend2* gene using CRISPR DESIGN TOOLS (Millipore Sigma) (gRNA1-antisense: AGTAGCAGGCTGCATAAGT GGG; gRNA2-sense: AGACCAGC CTTATTGACCA TGG). SgRNAs were synthesized by Sigma and microinjected with Cas9 protein into the pronucleus of C57BL/6JOlaHsd zygotes. Edited founders were identified by PCR with primers flanking the targeted region and HincII digest. PCR products were further purified and determined by Sanger sequencing. Five out of 17 F0 mice were identified to carry desired mutations, including two homozygous males and three heterozygous females. Each of them was crossed with wild-type C57BL/6JOlaHsd to eliminate possible off-target mutations to generate pure heterozygotes. Female heterozygotes (*Bend2*$^{+/D11}$) were crossed with wild-type males (*Bend2*$^{+/y}$) to obtain wild-type females (*Bend2*$^{+/+}$), heterozygous females (*Bend2*$^{+/D11}$), wild-type male (*Bend2*$^{+/y}$), and mutant males (*Bend2*$^{D11/y}$) offsprings. Female heterozygotes (*Bend2*$^{+/D11}$) were crossed with mutant males (*Bend2*$^{D11/y}$) to obtain mutant females (*Bend2*$^{D11/D11}$), heterozygous females (*Bend2*$^{+/D11}$), wild-type males (*Bend2*$^{+/y}$) and mutant males (*Bend2*$^{+/y}$). Mice from at least F2 generation were analyzed for their phenotyping. All experiments used at least three animals from each genotype (unless mentioned in the text). Mutant males were compared with their wild-type littermates. Testes from 2 to 6-months-old mice were collected and processed for adult mice. Female mutants were compared to wild-type mice from other litters of the same age and from animals of closely related parents.

For genotyping, genomic DNA was extracted from mouse tails by overnight incubation at 56°C in lysis buffer (0.1 M Tris-HCl pH 8.5–9, 0.2 M NaCl, 0.2% SDS, 5 mM EDTA, and 0.4 mg/ml proteinase K), followed by precipitation with isopropanol and washes in cold 70% ethanol and subjected to PCR using NZYTaq II 2 x Green Master Mix employing primer pair (forward: TTGCCAGTGGGGTATTACGA , and reverse: CTGGAAGGCAGGAAGTTTAACA). For female genotyping, the identified possible

homozygous BEND2 females were subjected to an extra PCR using the primer pair (forward: TTTG CTCCACTGTTTCACGC, and reverse: TCCCTTTAAACTGCCAACAACA) to confirm the homozygosity.

The experiments performed in this study complied with EU regulations and were approved by the Ethics Committee of the UAB and the Catalan Government (5322-CEEA-UAB).

## Ovarian stimulation and in vitro fertilization

Intra-peritoneal injection of 15 IU pregnant mare's serum gonadotrophin per female was administered to three *Bend2*$^{+/D11}$ and three *Bend2*$^{D11/D11}$ 7-month-old mice. After 49.5 hr, 10 IU human chorionic gonadotrophin (hCG) per female was injected and 13.5 hr later mice were sacrificed. Oviducts of treated females were dissected under stereo-microscope and cumulus masses were released into a drop of fertilization medium (0.25 mM GSH in HTF medium). Oocytes released from each female mice were counted. Time of ovulation was counted as 0.5 dpc.

For in vitro fertilization, fresh sperm from a proven fertile, C57BL/6 N wild-type male of 4 months of age were used and poured into a dish containing oocytes in fertilization medium. After 3 hr of fertilization, oocytes were washed and O/N cultured in HTF medium. Assessment of fertilized, unfertilized oocytes was done under stereo-microscope. From 2 cell-stage until blastocysts, embryos were cultured in KSOM medium.

## Total RNA purification and RT-PCR

Total RNA was purified from mouse tissues using the RNeasy Plus Mini Kit (Qiagen) and then transcribed into cDNA using the iScript cDNA Synthesis Kit (Bio-Rad), following the manufacturer's instructions. cDNA was amplified using NZYTaq II 2 x Green Master Mix with gene-specific primers (255P1 forward: TAGGGACCAAGAACCTGCTG, and reverse: TCCTGAAGCCACTGAGAAGG) and β-actin primers (forward: AGGTCTTTACGGATGTCAACG, and reverse: ATCTACGAGGGCTATGCTCTC) as control. A minus Reverse Transcription control (RT-) containing all the reaction components except the reverse transcriptase was included for testing for contaminating DNA.

## Gene cloning and transfection

cDNA from wild-type mouse testis was subject to PCR using Phusion High-Fidelity DNA Polymerase (Thermo Fisher Scientific) with a pair of specific primers (forward: agaaATGCCAGGAAAAACTGAAG, and reverse: TTAAGCTATTGCATTCCTTGGG for pEGFP-C1 vector and gAGCTATTGCATTCCTTGGG C for pEGFP-N1 vectors) targeting at both ends of the coding sequence to amplify the full-length of *255p1 (Bend2* novel splice variant). DNA purified from PCR reaction mix was phosphorylated and inserted into dephosphorylated pEGFP-C1 and pEGFP-N1 vectors. Plasmids were transformed to 5-alpha competent *E. coli* cells (NEB) and identified by colony PCR followed by Sanger sequencing.

pEGFP-C1-BEND2 or pEGFP-N1-BEND2 were transfected into the human embryonic kidney cell line (HEK 293T, obtained from the Cell Culture, Antibody Production and Cytometry Service (SCAC) from UAB, its identity was confirmed by STR profiling and were mycoplasma negative) by using jetPEI DNA transfection reagent (Polyplus Transfection) and into 16–18 dpp wild-type live mouse testis by electroporation (*Shibuya et al., 2014*). Mice were sacrificed 24–72 hr post-electroporation and used for experiments.

## Antibody generation

To generate *in-house* BEND2 polyclonal antibody, the full-length of 255p1 (Bend2 novel splice variant) was amplified from pEGFP-C1-BEND2 and cloned into modified bacterial expression vector pET-28a (+)-TEV. The recombinant His-tagged BEND2 protein was expressed in *E. coli* BL21(DE3) competent cells by IPTG induction and then purified by affinity chromatography using HisTrap HP column (Cytiva). Purified His-tagged BEND2 protein was treated with TEV protease, followed by another affinity purification, to remove the His-tag. His-tag removed BEND2 protein was used to immunize rabbits. After immunization, the immune response was checked by ELISA. Antibodies were purified from rabbit serum by affinity chromatography using Affi-Prep protein A resin cartridge (Bio-Rad).

## Western blot

Total protein was extracted from mouse testis using RIPA lysis buffer (50 mM Tris pH 8, 1% Triton X-100, 0.1% SD, 150 mM NaCl, 1 mM EDTA, 0.5% Sodium Deoxycholate, 10 mM NaF, 1x Protease

Inhibitor). Testis tissue was disrupted and homogenized thoroughly with a pestle in RIPA lysis buffer, followed by 10 min incubation at 95°C. Each lysate sample's protein concentration was determined using Pierce BCA Protein Assay Kit (Thermo Scientific). 50 µg of total protein was loaded per well, and samples were separated by TGX-PAGE gel electrophoresis in Tris-Glycine-SDS buffer. Proteins were transferred to PVDF membranes (Bio-Rad) by wet electroblotting in Tris/glycine buffer. Membranes were blocked in 5% non-fat milk in PBS for 2 h at room temperature, followed by incubation of primary antibody diluted in blocking buffer overnight at 4 °C. The next day, after three washes of PBST (PBS containing 0.05% Tween-20), membranes were incubated in anti-rabbit HPR conjugate antibody (1:5000, 170–6515, Bio-Rad) in PBS for 1 hr at room temperature. ECL substrate (Bio-Rad) was used for chemiluminescent detection. Imaging was performed on ChemiDoc Touch Imaging system (Bio-Rad), and images were analyzed by Imagelab software (Bio-Rad). Primary antibodies used in WB: anti-BEND2 Rb (*in-house*, 3.2 µg/ml), anti-Ku70 Rb (abcam, 1:2000), anti-LINE1 ORF1p Rb (abcam, 1:2000), and anti-GAPDH Rb (abcam, 1:2000).

## Nuclei spreading and immunofluorescence

Spermatocyte nuclei spreading was prepared using frozen testes. Protocol was adapted from *Liebe et al., 2004*: briefly, a small portion of testis was cut and minced thoroughly by a sterile blade in cold PBS (pH 7.4) containing 1x protease inhibitor, PI (Roche Diagnostics) on a petri dish; cell mixture was transferred to a sterile Eppendorf and sat for 15 min to sediment; 25 µl of supernatant cell suspension was spread onto a glass slide and incubated with 80 µl of 1% Lipsol containing 1x PI, for 15–20 min, followed by fixation in 150 µl of the PFA fixative solution (1% Paraformaldehyde pH 9.2–9.4, 15% Triton X-100, 1 x PI) for 2 hr at room temperature in a closed humid chamber. Oocyte nuclei spreading was prepared from fresh fetal and perinatal ovaries. Briefly, one pair of ovaries were dissected from each female under a stereomicroscope (Nikon SMZ-1); ovaries were first incubated in 500 µl of M2 medium (Sigma-Aldrich) containing 2.5 mg/ml collagenase (Sigma-Aldrich) for 30 min at 37°C, and then incubated in 500 µl of hypotonic buffer (30 mM Tris-HCl pH 8.2, 50 mM Sucrose, 17 mM Sodium Citrate, 5 mM EDTA, 0.5 mM DTT, 1 x PI) for 30 min at room temperature; finally, single-cell suspension was prepared by disaggregating ovaries in 100 mM sucrose by pipetting under the stereo microscope. Each 10 µl of the cell suspension was spread onto a glass slide, followed by fixation in 40 µl of the PFA fixative solution (1% PFA, 5 mM Sodium Borate, 0.15% Triton X-100, 3 mM DTT, 1 x PI, pH 9.2) for 2 hr at room temperature in a closed humid chamber. Slides were dried under a fume hood and then washed in 0.4% Photoflo (Kodak) solution four times.

For immunofluorescence staining, slides of spermatocyte or oocyte nuclei spreading were blocked in freshly made blocking solution (0.2% BSA, 0.2% gelatin, 0.05% Tween-20 in PBS) at room temperature, followed by incubation of primary antibody diluted in blocking solution in a humid chamber overnight at 4°C; the next day, slides were washed four times in blocking solution, and incubated in secondary antibody diluted in blocking solution in a humid chamber for 1 hr at 37°C, followed by another four washes in blocking solutions. Drained slides were mounted with 0.1 µg/ml DAPI in Vectashield antifade mounting medium and analyzed with an epifluorescence microscope (Zeiss Axioskop). Primary antibodies used for IF: anti-SYCP3 Ms or Rb (abcam, 1:200), anti-SYCP1 Rb (abcam, 1:200), anti-phospho-Histone H2A.X Ms (Millipore, 1:400), anti-MLH1 Ms (BD Biosciences, 1:50), RPA32 (4E4) Rat (cell signaling, 1:100), anti-RAD51 (ab-1) Rb (Millipore, 1:100), and anti-GFP Rb (Thermo Fisher Scientific, 1:200).

## PAS-Hematoxylin, immunohistochemical, and TUNEL staining

Fresh mouse tissues were fixed overnight at 4°C with freshly made 4% PFA (0.4 g paraformaldehyde dissolved in 10 ml PBS, pH 7.4) for immunohistochemical (IHC)/TUNEL assay or with Bouin's fixative for the use of PAS (Periodic Acid Schiff) staining. After fixation, tissues were washed in PBS, dehydrated in a series of ethanol with increasing concentration, cleared by histoclear and infiltrated by paraffin in sequence; embedded tissues were cut into thin slices (6–7 µm for testes; 4 µm for ovaries) using a microtome, which were mounted on poly-L-lysine coated slides and dry overnight at 37°C. Before any staining procedures, slides were deparaffinized in xylene and a series of ethanol with decreasing concentration in sequence and rehydrated in distilled water.

For IHC, an antigen retrieval step was performed to expose the epitopes masked by PFA fixation: incubating the slides in sodium citrate buffer (10 mM Sodium citrate, 0.05% Tween 20, pH 6.0) or

Tris-EDTA buffer (10 mM Tris, 1 mM EDTA, 0.05% Tween 20, pH 9.0) for 30 min at 95–100° C. Subsequently, an immunofluorescence was performed as described above and the used primary antibodies were anti-DDX4 (Abcam, 1:100) and anti-LINE-1 (Abcam, 1:100).

For TUNEL assay, tissue section samples were permeabilized in 0.5% Triton X-100 in PBS for 15 min followed by two washes in PBS, incubated in background reducing solution (Dako), followed by a rinse in PBS, incubated in TUNEL reaction mixture (10% TdT enzyme solution, Roche Diagnostics) for 1 hr at 37°C in a humid chamber, followed by three washes in PBS. Slides were mounted with 15 µl DAPI (0.1 µg/ml in Vectashield antifade mounting medium) and analyzed with an epifluorescence microscope (Zeiss Axioskop).

For PAS-Hematoxylin (PAS-H) staining, tissue sections were oxidized by 1% Periodic Acid solution for 10 min, followed by two washes in distilled water, stained by Schiff's reagent for 30 min in darkness, followed by two washes in sulfurous water (10% Potassium metabisulfite, 0.1 M HCl in Milli-Q water) and two washes in distilled water; counterstained in Mayer's Hematoxylin for 1 min and rinsed in running tap water; dehydrated in a series of ethanal with increasing concentrations and xylene in sequence; mounted with DPX mounting medium and analyzed with an optical microscope. Images were captured using a Zeiss Axioskop microscope.

## Follicle count and classification

The whole ovaries were sectioned for follicle quantification in newborn, young, adolescent, and adult mice. Eight consecutive sections were mounted on each slide. Every third section was counted per slide to avoid counting follicles more than once. Five alternate slides per animal were counted (both ovaries). Follicles were only counted if the nucleus of the oocyte was visible. They were classified as primordial if they contained one single layer of plane squamous granulosa cells or as primary if they displayed one layer of cuboidal granulosa cells. Furthermore, follicles containing both plane and cuboidal cells were classified based on the predominant granulosa cell morphology. Follicles were classified as secondary if they showed more than one layer of cuboidal granulosa cells with no visible antrum. All follicles displaying an antral cavity of any size were classified as antral. Counting and classification were performed under a bright-field Zeiss Axioskop microscope.

## Image processing, analysis, and statistical analysis

All the images were processed by Adobe Photoshop. Fluorescence intensity was quantified by ImageJ; fluorescence signals were counted manually or by ImageJ. Data analysis and statistical inference were performed using GraphPad Prism 8 software. The sample size was determined based on our previous experience analyzing other mutant phenotypes and the variability observed within each type of analysis (*Marcet-Ortega et al., 2017*; *Martínez-Marchal et al., 2020*; *Pacheco et al., 2018*; *Roig et al., 2010*; *Ruth et al., 2021*). Statistical methods and p-values are presented in each graph or in the main text or figure legends. Statistical significances were determined using t-tests for normally distributed or Mann-Whitney U-tests for non-normally distributed continuous data. One-way ANOVA test was used for analysis of homogeneity of variances. For all tests, statistical significance was considered for $p < 0.05$.

## Acknowledgements

This work was supported by Spanish Ministerio de Ciencia, Innovación e Universidades grants (I.R: PID 2019-107082RB-I00, PID2022-138905OB-I00; A.M.P.: PID2020–120326RB-I00), Junta de Castilla y León grants (A.M.P.: CSI017P23 and CSI017P23), and a grant from La Fundació Marató de TV3 (I.R.: 677 /U/2021). Yan Huang is a recipient of a fellowship from the China Scholarship Council (201607040048). Cristina Madrid-Sandín is the recipient of an FPU fellowship from the Spanish Ministerio de Ciencia, Innovación e Universidades (FPU19/02885). Nikoleta Nikou and Carolina Buza-Tarna are supported by an FPI fellowship from the Spanish Ministerio de Ciencia, Innovación e Universidades (PRE2020-094355 and PREP2022-000758, respectively). Maria López-Panadés is supported by a FI-SDUR fellowship from the Catalan Government (2023 FISDU 00058). Moreover, we would like to acknowledge the rest of the members of the Roig Lab for their support, comments, and discussions about the project.

# Additional information

## Funding

| Funder | Grant reference number | Author |
|---|---|---|
| Ministerio de Ciencia, Innovación y Universidades | PID2019-107082RB-I00 | Ignasi Roig |
| Ministerio de Ciencia, Innovación y Universidades | PID2022-138905OB-I00 | Ignasi Roig |
| Ministerio de Ciencia, Innovación y Universidades | PID2020-120326RB-I00 | Alberto M Pendás |
| Fundació la Marató de TV3 | 677/U/2021 | Ignasi Roig |
| China Scholarship Council | 201607040048 | Yan Huang |
| Ministerio de Ciencia, Innovación y Universidades | FPU19/02885 | Cristina Madrid-Sandín |
| Ministerio de Ciencia, Innovación y Universidades | PRE2020-094355 | Nikoleta Nikou |
| Ministerio de Ciencia, Innovación y Universidades | PREP2022-000758 | Carolina Buza |
| Agència de Gestió d'Ajuts Universitaris i de Recerca | 2023 FISDU 00058 | Maria López-Panadés |
| Junta de Castilla y León | CSI017P23 | Alberto M Pendás |

The funders had no role in study design, data collection and interpretation, or the decision to submit the work for publication.

## Author contributions

Yan Huang, Conceptualization, Data curation, Formal analysis, Investigation, Visualization, Methodology, Writing – original draft, Writing – review and editing; Nina Bucevic, Carmen Coves, Natalia Felipe-Medina, Marina Marcet-Ortega, Data curation, Formal analysis, Investigation, Visualization, Methodology; Nikoleta Nikou, Cristina Madrid-Sandín, Maria López-Panadés, Carolina Buza, Data curation, Formal analysis, Investigation, Visualization, Methodology, Writing – review and editing; Neus Ferrer Miralles, Conceptualization, Validation, Investigation, Visualization, Methodology, Writing – review and editing; Antoni Iborra, Conceptualization, Formal analysis, Validation, Investigation, Visualization, Methodology, Writing – review and editing; Anna Pujol, Conceptualization, Formal analysis, Investigation, Visualization, Methodology, Writing – review and editing; Alberto M Pendás, Conceptualization, Supervision, Methodology, Writing – review and editing; Ignasi Roig, Conceptualization, Data curation, Formal analysis, Supervision, Funding acquisition, Validation, Investigation, Methodology, Writing – original draft, Project administration, Writing – review and editing

## Author ORCIDs

Marina Marcet-Ortega ⓘ https://orcid.org/0000-0002-5756-1373
Nikoleta Nikou ⓘ https://orcid.org/0000-0002-4579-0849
Alberto M Pendás ⓘ https://orcid.org/0000-0001-9264-3721
Ignasi Roig ⓘ https://orcid.org/0000-0003-0313-3581

## Ethics

The experiments performed in this study complied with EU regulations and were approved by the Ethics Committee of the UAB and the Catalan Government (5322-CEEA-UAB).

Reviewer #1 (Public review): https://doi.org/10.7554/eLife.96052.4.sa1
Reviewer #2 (Public review): https://doi.org/10.7554/eLife.96052.4.sa2
Reviewer #3 (Public review): https://doi.org/10.7554/eLife.96052.4.sa3
Author response https://doi.org/10.7554/eLife.96052.4.sa4

# Additional files

## Supplementary files

Supplementary file 1. Testis and body weight (TW/BW) ratio of male mice.

MDAR checklist

## Data availability

All the data analyzed in this study are available within the manuscript and its supporting materials.

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
