## [Editor Report · eLife Assessment]

This study provides **valuable** information on a novel gene that regulates meiotic progression in both male and female meiosis. The evidence supporting the conclusions of the authors is **solid**. This study will be of interest to developmental and reproductive biologists.

---

## [Referee Report · Reviewer #1 (Public review)]

Summary:

In this manuscript, the authors investigate the role of BEND2, a novel regulator of meiosis, in both male and female fertility. Huang et al have created a mouse model where the full-length BEND2 transcript is depleted but the truncated BEND2 version remains. This mouse model is fertile, and the authors used it to study the role of BEND2 on both male and female meiosis. Overall, the full-length BEND2 appears dispensable for male meiosis. The more interesting phenotype was observed in females. Females exhibit a lower ovarian reserve suggesting that full-length BEND2 is involved in the establishment of the primordial follicle pool.

Strengths:

The authors generated a mouse model that enabled them to study the role of BEND2 in meiosis. The role of BEND2 in female fertility is novel and enhances our knowledge of genes involved in the establishment of the primordial follicle pool.

Weaknesses highlighted previously:

The manuscript extensively explores the role of BEND2 in male meiosis; however, a more interesting result was obtained from the study of female mice.

---

## [Referee Report · Reviewer #2 (Public review)]

In their manuscript entitled "BEND2 is a crucial player in oogenesis and reproductive aging", the authors present their findings that full-length BEND2 is important for repair of meiotic double strand break repair in spermatocytes, regulation of LINE-1 elements in spermatocytes, and proper oocyte meiosis and folliculogenesis in females. The manuscript utilizes an elegant system to specifically ablate the full-length form of BEND2 which has been historically difficult to study due to its location on the X chromosome and male sterility of global knockout animals.

The authors have been extremely responsive to reviewer critiques and have presented strong data and appropriate conclusions, making it an excellent addition to the field.

---

## [Referee Report · Reviewer #3 (Public review)]

Huang et al. investigated the phenotype of Bend2 mutant mice which expressed truncated isoform. Bend2 deletion in male showed fertility and this enabled them to analyze the BEND2 function in females. They showed that Bend2 deletion in females showed decreasing follicle number which may lead to loss of ovarian reserve.

Strengths:

They found the truncated isoform of Bend2 and the depletion of this isoform showed decreasing follicle number at birth.

Weaknesses highlighted previously:

The authors showed novel factors that impact ovarian reserve. Although the number of follicles and conception rate are reduced in mutant mice, the in vitro fertilization rate is normal and follicles remain at 40 weeks of age. It is difficult to know how critical this is when applied to the human case.

[Editors' note: We thank the authors for considering the previous recommendations and suggested corrections.]

---

## [Author Response]

The following is the authors’ response to the previous reviews.

**Public Reviews:**

**Reviewer #1 (Public review):**
Summary:In this manuscript, the authors investigate the role of BEND2, a novel regulator of meiosis, in both male and female fertility. Huang et al have created a mouse model where the full-length BEND2 transcript is depleted but the truncated BEND2 version remains. This mouse model is fertile, and the authors used it to study the role of BEND2 on both male and female meiosis. Overall, the full-length BEND2 appears dispensable for male meiosis. The more interesting phenotype was observed in females. Females exhibit a lower ovarian reserve suggesting that full-length BEND2 is involved in the establishment of the primordial follicle pool.Strengths:The authors generated a mouse model that enabled them to study the role of BEND2 in meiosis. The role of BEND2 in female fertility is novel and enhances our knowledge of genes involved in the establishment of the primordial follicle pool.Weaknesses:The manuscript extensively explores the role of BEND2 in male meiosis; however, a more interesting result was obtained from the study of female mice.

We sincerely appreciate the reviewer’s thoughtful evaluation of our work and recognition of the strengths of our study. We are especially grateful for the acknowledgment of the novelty of our findings regarding the role of BEND2 in female fertility. While we extensively characterized the e ects of BEND2 depletion in male meiosis, we agree that the phenotype observed in females provides particularly interesting insights into the establishment of the primordial follicle pool.

**Reviewer #2 (Public review):**
In their manuscript entitled "BEND2 is a crucial player in oogenesis and reproductive aging", the authors present their findings that full-length BEND2 is important for repair of meiotic double strand break repair in spermatocytes, regulation of LINE-1 elements in spermatocytes, and proper oocyte meiosis and folliculogenesis in females. The manuscript utilizes an elegant system to specifically ablate the full-length form of BEND2 which has been historically di icult to study due to its location on the X chromosome and male sterility of global knockout animals.The authors have been extremely responsive to reviewer critiques and have presented strong data and appropriate conclusions, making it an excellent addition to the field.

We are truly grateful for the reviewer’s thoughtful review and recognition of the key contributions of our study. We appreciate the acknowledgment of how our model overcomes the challenges in studying BEND2 and the importance of our findings in both male and female meiosis. We also value the reviewer’s encouraging comments on our responsiveness to their feedback and the quality of our data and conclusions.

**Reviewer #3 (Public review):**
Huang et al. investigated the phenotype of Bend2 mutant mice which expressed truncated isoform. Bend2 deletion in male showed fertility and this enabled them to analyze the BEND2 function in females. They showed that Bend2 deletion in females showed decreasing follicle number which may lead to loss of ovarian reserve.Strengths:They found the truncated isoform of Bend2 and the depletion of this isoform showed decreasing follicle number at birth.Weaknesses:The authors showed novel factors that impact ovarian reserve. Although the number of follicles and conception rate are reduced in mutant mice, the in vitro fertilization rate is normal and follicles remain at 40 weeks of age. It is difficult to know how critical this is when applied to the human case.

We greatly appreciate the reviewer’s comments and recognition of the strengths of our work. We are grateful for their acknowledgment of our findings related to the truncated isoform of Bend2 and its e ect on ovarian reserve. We also agree that, although our study provides important insights, we are still far from directly applying these results to human clinical scenarios. There is much further research needed before these findings can be translated.

**Recommendations for the authors:**

**Reviewer #1 (Recommendations for the authors)::**
The authors have addressed all concerns both editorially and experimentally. This is a very nice manuscript, and I congratulate the authors on their work.

We sincerely appreciate your kind words and thoughtful review. Your feedback has been invaluable in improving our manuscript, and we are grateful for your time and effort. Thank you for your support and encouragement!

**Reviewer #2 (Recommendations for the authors)::**
In Figure 3, graphs in panels C & D have typos in the early zygotene column where it reads "zyotene".

We appreciate your careful review and for pointing out the typos in Figure 4, which has been corrected in the new version of the manuscript.

**Reviewer #3 (Recommendations for the authors):**
・Since there are two isoforms of Bend2, and the authors depleted one isoform, this is not suitable to use "full length" in the titles and in the manuscripts.

We respectfully disagree with the reviewer’s comment. In our mouse model, we specifically remove the full-length isoform of Bend2. Therefore, we consider it appropriate to refer to it as such in the manuscript. Our results indicate that the full-length isoform is not required to complete meiotic prophase in males but is indispensable for setting up the ovarian reserve in females. We appreciate the reviewer’s input and are happy to clarify this point further if needed.

・Is there any reason why authors used 7 month old females for in vitro fertilization? It may not be recognized as aged mice but it seems a bit old to perform IVF especially when the ovarian reserve in mutant mice is decreased. If there is any reason, please clarify it. In addition, since the authors added IVF data, which showed similar fertilization ratio between control and mutant, the authors need to discuss why the litter size was decreased in mutant mice. It may be to strong to conclude "subfertility".

We used 7-month-old females for IVF because this falls within the age range of the samples analyzed for ovarian reserve, with the oldest females being 8 months old. Regarding the apparent discrepancy between IVF results and litter size, we addressed this in the discussion section of the manuscript: 'Interestingly, our mutant oocyte quality analysis suggests that mature oocytes from mutant females are equally competent to develop into a blastocyst as control ones. These data suggest that the subfertility observed in Bend2 mutants may be due to errors in later developmental stages, such as implantation or organogenesis.' We appreciate the reviewer’s feedback and hope this clarification helps.